# Biomimetic Whitening Effect of Polyphosphate-Bleaching Agents on Dental Enamel

**DOI:** 10.3390/biomimetics7040183

**Published:** 2022-10-29

**Authors:** Abdullah Alshehri, Basil Almutairi, Carlos A. Jurado, Kelvin I. Afrashtehfar, Shug Albarrak, Asma Alharbi, Alanoud Alenazi, Hamid Nurrohman, Abdulrahman Alshabib

**Affiliations:** 1Department of Conservative Dental Sciences, College of Dentistry, Prince Sattam bin Abdulaziz University, Al-Kharj 11942, Saudi Arabia; 2Department of Restorative Dentistry, Division of Operative Dentistry, King Saud University College of Dentistry, Riyadh 11545, Saudi Arabia; 3Woody L. Hunt School of Dental Medicine, Texas Tech University Health Sciences Center, El Paso, TX 79905, USA; 4Clinical Sciences Department, College of Dentistry, Ajman University, Ajman City P.O. Box 346, United Arab Emirates; 5Department of Reconstructive Dentistry & Gerodontology, School of Dental Medicine, University of Bern, 3010 Berne, Switzerland; 6Dental Interns, College of Dentistry, King Saud bin Abdelaziz University for Health Science, Riyadh 11545, Saudi Arabia; 7Restorative Dentistry, A.T. Still University Missouri School of Dentistry & Oral Health, Kirksville, MO 63501, USA; 8Engineer Abdullah Bugshan Research Chair for Dental and Oral Rehabilitation, King Saud University, Riyadh 11545, Saudi Arabia

**Keywords:** CIELAB, color stability, enamel, hypersensitivity, tooth whitening

## Abstract

This in vitro study investigated the extrinsic tooth-whitening effect of bleaching products containing polyphosphates on the dental enamel surface compared to 10% carbamide peroxide (CP). Eighty human molars were randomly allocated into four whitening-products groups. Group A (control) was treated with 10% CP (Opalescence). The other groups with non-CP over-the-counter (OTC) products were group B = polyphosphates (iWhiteWhitening-Kit); group C = polyphosphates+fluoride (iWhite-toothpaste); and group D = sodium bicarbonate (24K-Whitening-Pen). *L**, *a**, *b** color-parameters were spectrophotometer-recorded at baseline (T0), one day (T1), and one month (T2) post-treatment. Changes in teeth color (ΔE_ab_) were calculated. Data were analyzed using ANOVA and the Bonferroni test (α = 0.05). Groups A, B, and D showed significant differences in Δ*L**&Δ*a** parameters at T1, but not in Δ*b** at T0. Group C showed no difference for Δ*L**, Δ*a**, Δ*b** at T0 and T1. Group A showed differences for Δ*L**, Δ*a**, Δ*b**, at T2, while groups B, C, and D had no difference in any parameters at T0. At T1, ΔE_ab_ values = A > D> B > C (ΔE_ab_ = 13.4 > 2.4 > 2.1 > 1.2). At T2, ΔE_ab_ values increased = A > B > C > D (ΔE_ab_ = 12.2 > 10.6 > 9.2 > 2.4). In conclusion, the 10% CP and Biomimetic polyphosphate extrinsic whitening kit demonstrated the highest color change, while simulated brushing with dark stain toothpaste and a whitening pen demonstrated the lowest color change at both measurement intervals.

## 1. Introduction

Tooth whitening has become increasingly popular in recent years due to the convenience with which it may be accomplished and the positive impact it has on smile aesthetics and overall quality of life [1]. Thus, tooth whitening is now considered one of the most desired dental treatments among the general population [1,2]. In recent years, several tooth whitening products have been introduced and advertised by various manufacturers, which has significantly increased the value of the global oral health care market [2,3].

Professional tooth whitening procedures are performed either in-office by a dental professional or at home under supervision, both of which are accomplished utilizing various techniques and application regimens [4,5]. Traditionally, the two most common agents in tooth whitening products are hydrogen peroxide (HP) and carbamide peroxide (CP), which are used at various percentages (HP-30–35%; CP-10–20%) depending on the clinical situation or the procedure [6]. However, the whitening effect of chemically induced in-office tooth whitening is primarily attributed to the action of hydrogen peroxide [7,8].

With an increase in demand for tooth whitening, many non-supervised over-the-counter (OTC) whitening products are readily available in the market for self-application by individuals. These products are available as rinses, toothpaste, whitening strips, gels, chewing gums, or paint-on brushes. They usually contain a low percentage of CP, HP, or other alternative active ingredients [9,10]. Furthermore, OTC bleaching products have been linked to oral infections, burning sensation, compression of the gums, and nerve and tooth enamel damage [11]. Previous studies evaluating the whitening effect of OTC products have low or similar effects compared to professionally applied products [11,12,13]. This justifies that not all OTC tooth whitening products are effective as the professionally applied and supervised whitening procedures [14,15].

The American Dental Association (ADA) recommends consulting with a licensed dentist to determine whether tooth whitening is an appropriate treatment, in order to educate the public. Furthermore, some OTC whitening products bear the ADA “Seal of Acceptance”, indicating that they are safe and effective when used as directed [16,17]. OTC whitening products are a growing industry with a billion-dollar global market. Many OTC products have been introduced in recent years by major consumer outlets. These products’ lack of data and easy availability is concerning, as it may potentially harm the customers’ dentition. Because of their ‘Do-It-Yourself’ nature, over-the-counter products also pose the risk of mishandling, overuse, and abuse [18].

A high peroxide concentration in whitening products has been associated with hypersensitivity [19]. As a result, many oral health regulatory agencies in the United States, the United Kingdom, and Europe have restricted the concentration of peroxides in over-the-counter (OTC) products to levels much lower than those recommended for vital tooth bleaching. Hence, this has paved the development of ‘non-hydrogen peroxide’ agents to be incorporated in the OTC products [2], such as biomimetic sodium hexametaphosphate, sodium fluoride, and sodium bicarbonate. However, there is limited research or clinical data regarding their whitening effect and safety.

Therefore, the purpose of this study was to compare the efficacy of some of the newly introduced OTC tooth whitening products to a reliable tooth whitening protocol in terms of color stability. The study hypothesis was that there is no difference in *L**, *a**, *b** color parameters, nor the $\Delta E_{ab}$, among the three OTC tooth whitening products when compared to a professional tooth whitening product.

## 2. Materials and Methods

### 2.1. Specimen Preparation

An ethical approval application (PSAU2021013) submitted to the deanship of scientific research at Prince Sattam Bin Abdulalziz University was approved and granted an exemption. Eighty sound human molars extracted for therapeutic purposes were collected and stored in thymol solution until use. The roots of all the teeth were separated from the crown portion using a diamond saw in a low-speed precision cutting machine (IsoMet 1000, Buehler, Lake Bluff, Illinois). All the teeth were thoroughly cleaned using plain pumice (Preppies, WhipMix, Louisville, KY, USA) and prophy cups (Young Innovations Inc, Algonquin, IL, USA), and individually embedded in 6 mm thick polyvinyl chloride (PVC) molds using clear self-cure orthodontic resin (Techno Sin Standard Kit, Protechno, Girona, Spain), leaving the outer enamel surfaces uncovered by the resin. Then, the buccal surface of each sample was polished using silicon carbide sandpaper with different grit sizes (400, 600, 1200, and 2400) attached to an automatic polishing machine (Rotopol-V, Struers, Cleveland, OH, USA). Following polishing, a standardized treatment area (6 mm in diameter) was established on the buccal surface (6 mm in diameter) using nail varnish. All samples were stored in distilled water prior to the experiment. The prepared specimens were randomly distributed into four groups (*n* = 20) according to the whitening products tested (Table 1).

Group A: Teeth specimens were treated with 10% carbamide peroxide (Opalescence, Ultradent, UT, USA) using a custom-made suck-down tray. Teeth were treated for 8 h per day for 10 days and served as the positive control. Group B: Specimens were treated with iWhite instant Dark Stains Whitening kit (Sylphar, Deurle, Belgium) for 20 min a day for 5 days using a prefabricated tray. Group C: Specimens were subjected to brushing using iWhite instant Dark Stains toothpaste in a toothbrushing simulator (ZM 3, SD Mechatronik GMBH, Feldkirchen-Westerham, Germany). The simulator machine was equipped with 12 individual slots to which to attach twelve toothbrushes. The mounted specimens were positioned inside the containers and secured using dental putty (3M™ Express™ Standard Putty Kit, New South Wales, Australia). Toothbrushes with soft straight bristles (TARA Toothbrush Company LLC, Dammam, Saudi Arabia) were engaged to the simulator brush slots. The container was filled with a slurry of dentifrice (iWhite instant Dark stains toothpaste, Sylphar, Deurle, Belgium) and distilled water at a ratio of 1:2 by weight. Tooth brushing was carried out with a forward and backward movement under a load of 2N, stroke rate of 160 cycles/minute, and travel length of 38 mm in order to cover the entire specimen surface. The total brushing time was 5 min, with 900 brushing cycles, simulating 60 days of tooth brushing. The brushing time was calculated in accordance with previous studies [20,21], which state that the maximum contact duration per tooth surface per day is 5 s with a twice-daily brushing habit. Group D: The specimen surface was treated with a whitening pen (24K White Charcoal Teeth Whitening Pen, Active Wow, Tallahassee, FL, USA). The pen was swiped up and down to spread the gel onto the specimen surface and was allowed to dry for 60 s. The specimen was rinsed under running water after 20 min. The application process was performed once daily for 2 days. All the specimens were stored in distilled water in an incubator throughout the experiment, except during color measurements.

### 2.2. Color Measurement

The color of the specimens was recorded using International Commission on Illumination (CIE) *L**, *a**, *b** parameters at baseline (T0), one day (T1), and one month (T2) following the treatment regimen using a visible UV light spectrophotometer (LabScan^®^ XE, Hunter Associates Laboratory Inc., VA, USA). On *L***a***b** coordinates, the CIE system is a chromatic value color space that measures both value and chroma: *L** is the color’s lightness (100: white; 0: black); *a** is the color’s red (*a** > 0) and green (*a**< 0) dimension; and *b** is the color’s yellow (*b** > 0) and blue (*b** < 0) dimension. [22] The color measurement device was calibrated according to the manufacturer’s guidelines before taking the readings. Three readings were obtained for each specimen, and the average was calculated.

### 2.3. Color Change

The color change (ΔE_ab_) of the specimens at T1 and T2 relative to baseline (T0) was calculated using the ISO/TR-28642:2016 CIE-76 formula (Equations (1) and (2), respectively) [23,24].
ΔE_ab_ *(T1) = [(ΔL1-L0*)^2^ + (Δa1-A0*)^2^ + (Δb1-b0*)^2^]^½^(1)
 ΔE_ab_ *(T2) = [(ΔL2-L0*)^2^ + (Δa2-A0*)^2^ + (Δb2-b0*)^2^]^½^(2)

In this study, the 50:50% perceptibility threshold (PT) ∆E_ab_ value was determined to be ≤ 1.2, the while 50:50% acceptability threshold (AT) ∆E_ab_ value was 1.2–2.7 [25,26]. The ∆E_ab_ value above the AT is clinically unacceptable.

### 2.4. Data Analysis

Data were coded and entered using the statistical package for the social sciences (SPSS) package (version 22, IBM SPSS Inc., Chicago, IL, USA). Quantitative variables were summarized as mean and standard deviation. A Kolmogorov Smirnov test revealed a normal distribution of the data. The difference in the mean ∆E_ab_ among the groups was tested using analysis of variance (ANOVA). Bonferroni post-hoc analysis was used for comparison between the groups when the ANOVA test was significant (α − 0.05).

## 3. Results

### 3.1. L^*^, a^*^, b^*^ Parameters

The mean and standard deviations of the **L**, *a**, and *b*** color parameters at different measurement intervals for the study groups are presented in Table 2. The difference in lightness (Δ*L**) was in the range of 66–80.6, −1.4 to −4 for the red-green parameter (Δ*a**), and 3.3 to 5.7 for the yellow-blue parameter (Δ*b**). Relative to the baseline (T0) color parameters, all groups demonstrated increased Δ*L** values at T1 and decreased Δ*L** values at T2. The Δ*a** mean values increased at T1 except for group A specimens, and at T2, Δ*a** mean values decreased except for group B specimens, which showed no changes in the values. Similarly, Δ*b** decreased at T1 for the groups except for group B specimens, which showed slightly increased values; however, at T2, all the groups showed increased Δ*b** values.

For the statistical difference in the color parameter among the groups at T0, a non-significant difference was observed (*p* < 0.001). At T1 and T2, the differences in *L** and *a** color parameters were statistically different among the groups (*p* = 0.0001). However, no significant color change was observed for the *b** color parameter among the groups at T1 (*p* = 0.89) and T2 (*p* = 0.81) (Table 2).

The mean comparison and significance level of the color parameters (*L** *a** and *b**) at different measurement intervals is presented in Table 3. The difference in color parameters from T1-T0 was non-significant for the Δ*b** parameter in every study group, and Δ*a** for group D only (*p* > 0.05). The difference in *L**, *a**, and *b** color parameters from T2-T0 showed that all color parameters were statistically significant for groups A and B. On the contrary, the *L**, *a**, and *b**color parameters were statistically non-significant for groups C and D, but not for Δ*a** (*p* = 0.05) or Δ*b** (*p* = 0.04) in group D.

### 3.2. Color Difference

Figure 1 illustrates the mean ∆E_ab_ values of the study groups at different measurement times. At T1, ΔE_ab_ values were high for group A, followed by group D and group B (ΔE_ab_ = 13.4 > 10.6 > 9.2). The lowest ΔE_ab_ values were observed with group D (ΔE_ab_ = 2.4). At T2, group A demonstrated high ΔE_ab_ values, followed by group B and group C (ΔE_ab_ =13.4 > 2.4 > 2.1). The lowest ΔE_ab_ values were observed with group D (ΔE_ab_ = 1.2).

Changes in optical parameters such as color coordinates and chroma over the white background after 5000 cycles of thermocycling are listed in Table 3. The change in lightness (*L**) was in the range of −2.4 to 1.3, −0.3 to 2.1 for the red–green parameter (*a**), and −2.0 to 4.5 for the yellow–blue parameter. The ΔE_ab_ of the specimens induced by whitening materials were above the AT (ΔE_ab_ > 2.7) limit at T1, except for group C specimens (ΔE_ab_ = 1.2 −2.7). On the contrary, the specimens at T2 demonstrated ΔE_ab_ values below AT, except for 10% CP treated (control) specimens (ΔE_ab_ > 2.7). Group D specimens at T2 demonstrated ΔE_ab_ values equaling the maximum limit of PT (ΔE_ab_ = 1.2). The specimen groups showed significant color changes from T1 to T2 (*p* < 0.05), whereas the group C specimens demonstrated no significant color changes from T1-T2 (*p* = 0.23) (Figure 1).

## 4. Discussion

Tooth whitening procedures have become the most conservative and popular procedure to whiten discolored teeth. Consequently, many authors have focused their studies on determining the best clinical approach for tooth whitening with minimal side effects [27,28,29,30,31,32]. There are many commercially available whitening products (i.e., over-the counter, at-home, and in-office bleaching) that have yielded successful outcomes [27,31,33,34]. However, the lack of research related to the OTC products could affect the outcome when compared to the supervised use of a 10% CP home whitening regimen [35], which is considered the gold standard for tooth whitening [30,31,36].

The aim of the current study was to compare the efficacy of three recently available OTC tooth whitening products with the gold standard tooth whitening protocol [27,28,29,30,31,32]. There were significant differences among the *L** and *a** color parameters, but not for the *b** parameter, for all the tested groups. Furthermore, the color difference varied significantly between the tooth whitening products at different measurement times. The outcome of this study suggests partial rejection of the null hypothesis.

In this study, three OTC teeth whitening products were tested, which showed better whitening effects compared to the control 10% CP group (Figure 1). This could be attributed to the different active tooth whitening ingredients in these products. Unlike peroxide-based products, which may induce side effects (e.g., bleaching sensitivity and damage of the organic matrix of enamel and dentin), biomimetic polyphosphates and fluoride can be used on a daily basis without any adverse reaction. For group B and C, the products contained sodium hexametaphosphate and sodium fluoride. Both sodium hexametaphosphate and fluoride seem to be related to the formation of a “barrier” on the enamel surface against mineral loss. They also remineralize damaged enamel and strengthen the teeth structure. This will lead to the teeth regaining their natural white color, as well as prevention of tooth sensitivity. In addition, the consistency and application method could have also affected the outcome [37,38]. CP can be evenly applied on the tooth due to its gel-like consistency. Furthermore, the customized tray minimizes unnecessary contact of the agent with the gingiva, and protects the agent from saliva and lip movement. The toothpaste application contacts the tooth and is diluted by saliva flow [30]. However, it is difficult to apply the toothpaste evenly on the tooth surface, especially because of the differences in the toothbrushes’ bristles and shapes, brushing techniques, and the amount of toothpaste applied. The pen was the easiest product to apply, as it was designed to simply paint onto the teeth surfaces with a predetermined amount.

This study followed the manufactures’ instructions for each OTC product (Table 1). A previous study established that the concentration and contact time of the tooth whitening agent to the tooth was found to be crucial for the whitening outcome [39]. However, another study found the contact time of the whitening agent to be more important than the concentration [27]. The contact time of the OTC tooth whitening agents used in our study varied, and is definitely considered a short period of time when compared to the positive control group (eight hours per day for 10 days). Hence, 10% CP had the longest contact time with teeth, and, therefore, produced the highest color change at one day and at one month post treatment. However, the color change observed was above the AT. On the contrary, simulated brushing with Dark Stains toothpaste demonstrated the least color change, which was below the AT limit at both measurement intervals. The whitening effect of Dark Stains toothpaste can be attributed to the presence of hydrated silica and the continuous mechanical action of toothbrush bristles.

The current study emphasized enamel color changes, despite the color of subsurface dentin. A study showed that color changes in enamel contributed the most to the overall tooth color change when the buccal surface of the tooth was exposed to tooth whitening products [40]. However, it is not known if a whitening pen, which is activated by sodium bicarbonate, will have any tooth whitening impact to a greater extent beyond enamel layers. Our study showed that “Active Wow Pen,” which contains sodium bicarbonate and organic coconut charcoal gel, had a quick temporary tooth whitening effect. The color changes with gel applied through the pen demonstrated ΔE_ab_ values below the PT at one month post-treatment. Calatayud et al. [37] observed effective tooth color changes using the paint-on varnish containing 6% HP, and concluded that paint-on gels showed significant clinical efficacy, which is attributed to the active agent used. A review of the literature [10] concluded that paint-on gels at a higher percentage of HP were more effective than those at lower percentages, and that three times daily application was more effective than twice daily. Although few studies reported the effectiveness of paint-on gels, paint-on products vary in their chemical composition and application instructions. This makes it difficult to generalize the results to affirm them as an effective tooth whitening product.

It is well known that the visual color evaluation using the VITA shade guide is the most common method used. However, visual evaluation of tooth color can be influenced by various factors, such as the light source, the color of the gingiva, angle of measurement, skill of the examiner, and eye fatigue [41,42]. Spectrophotometers are considered more objective, reliable, and reproducible color evaluation instruments, especially when evaluating the effectiveness of a tooth whitening agent. However, reflections of the tooth surface, diameter and direction of the spectrophotometer tip, edge loss, and the background color of the specimen can affect the accuracy of color measurements. The research team followed the manufactures’ instructions for the spectrophotometer set-up and calibration. Furthermore, error range was minimized by taking repeated measurements.

Accurate interpretation of the results is the key to bridging the gap between in vitro research and clinical practice. The 50:50% perceptibility threshold is a standardized value at which the color difference is considered perceivable to the human eye by 50% of the population [26]. A color difference that is equal to or above ($\Delta E_{\mathrm{ab}}\geq 1.2$) is considered perceivable by the eye [25]. Our results showed that all teeth whitening products produced immediate results in which the color change was considered perceivable by the eye. Furthermore, when we compared color changes at one month (T2), we found that all color change values were also perceivable, except for specimens treated with the whitening pen (Δ*E*_ab_ = 1.2).

The present study had a few drawbacks. The application of the whitening products followed the manufacturer’s recommendation; however, it was under controlled laboratory conditions. During teeth whitening using OTC products, the lips, cheek, tongue, and saliva could reduce the product contact with the teeth. However, in the present study, the authors could not replicate those clinical situations. The effect of toothbrushing, food colors, beverages, and saliva during product use could have contributed to different color values compared to the present study’s outcome. The application of whitening products was performed by a dental clinician who had precise knowledge about the application. This may be different in real-life scenarios, when whitening is performed by the common public and could yield a different outcome. Finally, the specimens selected in this study were sound and free from stains. However, this cannot be controlled in a clinical environment and may produce a variation in the obtained color. Future studies should be directed toward evaluating the effects of these new OTC products on enamel hardness, [43,44] surface characteristics, and biosafety. Furthermore, the whitening effect of these new OTC tooth whitening products should be compared to those that obtained the ADA seal of acceptance. Another methodological limitation is that CIELAB was reported only as the parameter for color instead of reporting both CIELAB and DE2000. Nonetheless, many publications still report the CIELAB system only for color evaluation [45,46].

## 5. Conclusions

Based on the methodological approach and outcome of the statistical analysis, it was concluded that the 10% CP and simulated brushing with Dark Stain toothpaste demonstrated the highest and lowest color change at both measurement intervals. The color change using a whitening pen was below the PT limit, whereas all other study groups at all measurement intervals demonstrated Δ*E_ab_* values above the PT. The effectiveness and safety of OTC whitening products should be confirmed so that the public can make an informed decision when purchasing or using such products.

## Figures and Tables

**Figure 1 biomimetics-07-00183-f001:**
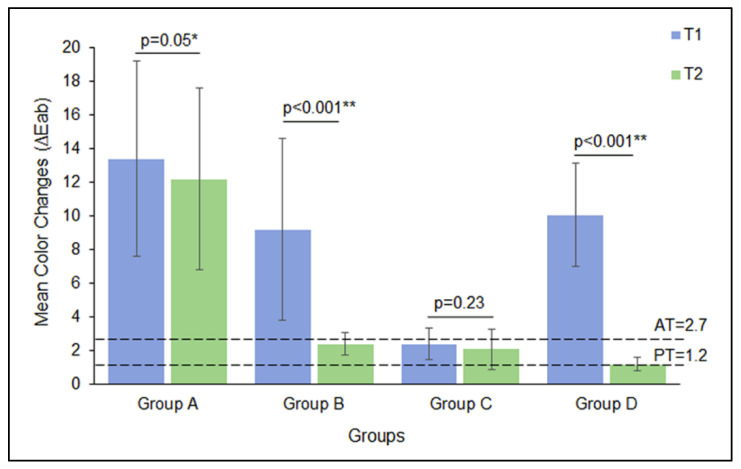
Mean ∆E_ab_ values of the study groups. Bar indicates standard deviation (SD), dashed line represents perceptibility threshold (PT) limit and acceptability threshold (AT) limit. * indicates significant difference in mean ∆E_ab_ between T1 and T2 within the groups (*p* < 0.05).

**Table 1 biomimetics-07-00183-t001:** Teeth whitening products tested in the study.

Group Code/ Tooth Whitening Agents/Manufacturers	Composition	Delivery Method/ Duration of Use
Group A Opalescence™ PF/ Ultradent, South Jordan, UT, USA	10% Carbamide Peroxide, 0.5% Potassium Nitrate, and 0.11% Fluoride Ions	Custom-made trays/ Eight hours per day for 10 days
Group B iWhite Dark Stains Whitening Kit/ Sylphar, Deurle, Belgium	Hydrated Silica, Sodium Hexametaphosphate, Mannitol, Chondrus Crispus Powder, Charcoal Powder.	Ready to use trays/ 20 min per day for 5 days
Group C iWhite Dark Stains Toothpaste/ Sylphar, Deurle, Belgium	Hydrated Silica, Sodium Hexametaphosphate, Mannitol, Chondrus Crispus Powder, Charcoal Powder, Sodium Fluoride, Sodium Saccharin.	Toothbrush/ Two brushing cycles per day for 60 days
Group D 24 K White Charcoal Teeth Whitening Pen/ Active Wow, Tallahassee, FL, USA	Sodium Bicarbonate, Carbomer, Polysorbate 20, Sodium Hydroxide, Potassium Sorbate, Organic Coconut Charcoal.	Whitening Pen/One application per day for two days

**Table 2 biomimetics-07-00183-t002:** Mean and standard deviation (SD) of the color parameters (*L**, *a** and *b**) at different measurement intervals.

Color Parameters	Mean ± SD			
	Group A	Group B	Group C	Group D
**T0**				
**Δ*L****	66 ± 3.7	71.5 ± 5.52	68.9 ± 4.36	70.2 ± 5.7
**Δ*a****	−2.8 ± 1.41	−3.2 ± 0.80	−3.5 ± 0.67	−3.6 ± 0.85
**Δ*b****	4.4 ± 4.21	4.1 ± 4.48	3.9 ± 4.1	4.2 ± 4.79
**T1**				
**Δ*L****	78.6 ± 7.17	80 ± 5	70 ± 4.26	80.6 ± 5
**Δ*a****	−1.6 ± 0.93	−3.5 ± 0.82	−3.7 ± 0.66	−4 ± 0.62
**Δ*b****	4.3 ± 2.93	4.2 ± 4.69	3.3 ± 4.59	3.6 ± 4.93
**T2**				
**Δ*L****	77.4 ± 6.6	73.7 ± 5.31	69.3 ± 3.85	70.9 ± 5.43
**Δ*a****	−1.4 ± 0.84	−3.5 ± 0.73	−3.4 ± 0.74	−3–3 ± 0.87
**Δ*b****	5.2 ± 3.12	4.8 ± 4.65	4.4 ± 4.78	5.7 ± 4.92

Group A—Opalescence™ PF; Group B—iWhite Dark Stains Whitening Kit; Group C—iWhite Dark Stains Toothpaste; Group D—24K White Charcoal Teeth Whitening Pen. * Indicates statistically significant difference between the whitening groups (*p* < 0.05).

**Table 3 biomimetics-07-00183-t003:** Mean comparison of the color parameters (*L**, *a**, and *b**) of the study groups at different measurement intervals.

Color Parameters	T1-T0	*p* Value	T2-T0	*p* Value
**Group A**				
**Δ*L****	12.6 ± 5.9	0.0001 *	11.4 ± 5.51	0.0001 *
**Δ*a****	3.7 ± 1.16	0.0001 *	−3.5 ± 1.20	0.001 *
**Δ*b****	−0.1 ± 1.89	0.61	0.8 ± 1.9	0.04 *
**Group B**				
**Δ*L****	8.6 ± 5.81	0.0001 *	2.2 ± 0.70	0.004 *
**Δ*a****	−0.4 ± 0.52	0.003 *	−0.3 ± 0.19	0.003 *
**Δ*b****	0.0 ± 2.80	1.0	0.7 ± 0.56	0.013 *
**Group C**				
**Δ*L****	1.2 ± 1.08	0.05 *	0.4 ± 1.38	1.0
**Δ*a****	−0.3 ± 0.2	0.05 *	0.1 ± 0.39	0.46
**Δ*b****	−0.6 ± 1.9	0.06	0.5 ± 1.90	0.34
**Group D**				
**Δ*L****	10.3 ± 3.06	0.0001 *	0.7 ± 0.79	0.08
**Δ*a****	−0.4 ± 0.38	0.08	0.3 ± 0.23	0.05 *
**Δ*b****	−0.6 ± 1.26	0.058	1.5 ± 0.45	0.04 *

Group A—Opalescence™ PF; Group B—iWhite Dark Stains Whitening Kit; Group C—iWhite Dark Stains Toothpaste; Group D—24K White Charcoal Teeth Whitening Pen. * Indicates statistically significant difference between the measurement intervals (*p* < 0.05).

## Data Availability

Not applicable.

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
