# Peer review of "Biomimetic Whitening Effect of Polyphosphate-Bleaching Agents on Dental Enamel"

_biomimetics, 2022, doi:10.3390/biomimetics7040183_

Round 1

Reviewer 1 Report

I think there are some minor and some major issues with the design of the manuscript entitled "Biomimetic Tooth-natural Whitening Effect of Polyphosphate Containing Bleaching Products" 

Line 44- please fill the reference missing

Line 77 and 83- please fill the reference missing

A major issue is why the authors used DE76 and not the newer formula DE2020? What was the threshold that they have chosen?

Line 97- how the teeth were cleaned?

Line 100- it is unclear that the word specimen is referring to the teeth. Did the authors use the polished teeth as specimens?Which part was used as specimen? the buccal side? how many mm was the specimen?

How the specimens were mounted for the spectrophotometer measurments? How the authors standardised the procedure?

Remove from all the tables the p value, you mention it to the manuscript so there is no need to state in the tables

Line 242- please fill the reference

Please insert a paragraph about the mechanism of tooth whitening of the products you mention (besides the CP)

Line 270- enamel color changes: please rephrase enamel has no color

Line 301- the threshold about DE should be mention also to the M&M section

Author Response

Author's Reply to the Review Report (Reviewer 1)

I think there are some minor and some major issues with the design of the manuscript entitled "Biomimetic Tooth-natural Whitening Effect of Polyphosphate Containing Bleaching Products" 

Line 44- please fill the reference missing

Thank you for your comment, change was made.

Line 77 and 83- please fill the reference missing

Thank you for your comment, change was made.

A major issue is why the authors used DE76 and not the newer formula DE2020? What was the threshold that they have chosen?

Thank you for your comment, we used CIE-76 formula as we are more familiar with this equation and has more relevance to literature.

Our statistical calculations were made based on a threshold of 1.2 difference in groups. It was added to the M&M section

Line 97- how the teeth were cleaned?

Thank you for your comment, change was made “All the teeth were cleaned Teeth were thoroughly cleaned using plain pumice (Preppies, Whipmix, USA ) and prophy cups (Young, USA)”

Line 100- it is unclear that the word specimen is referring to the teeth. Did the authors use the polished teeth as specimens?Which part was used as specimen? the buccal side? how many mm was the specimen?

Thank you for your comment, change was made “All the teeth were cleaned Teeth were thoroughly cleaned using plain pumice (Preppies, Whipmix, USA ) and prophy cups (Young, USA) and individually embedded in 6 mm thick polyvinyl chlorite (PVC) molds using clear self-cure orthodontic resin (Techno Sin Standard Kit, Protechno, Girona, Spain) leaving the outer enamel surfaces uncovered by the resin. Then, the buccal surface of each samplewas polished using silicon carbide sandpaper with different grit sizes (400, 600, 1200, and 2400) attached to an automatic polishing machine (Rotopol-V, Struers, Cleveland, OH, USA)”

How were the specimens mounted for the spectrophotometer measurements? How the authors standardized the procedure?

Thank you for your comment, change was made “ Following polishing, A standardized treatment area of 6mm in diameter was established on the buccal surface (6mm in diameter) using a nail varnish, all the specimens samples were stored in distilled water prior to the experiment”

Remove from all the tables the p value, you mention it to the manuscript so there is no need to state in the tables

Thank you for your comment, change was made

Line 242- please fill the reference

Thank you for your comment, references were included.

Please insert a paragraph about the mechanism of tooth whitening of the products you mention (besides the CP)

Thank you for your comment, article has been improved.

Line 270- enamel color changes: please rephrase enamel has no color

Thank you for your comment, change was made

Line 301- the threshold about DE should be mentioned also to the M&M section

Thank you for your comment, change was made

Reviewer 2 Report

After critically reviewing this Research Article titled "Biomimetic Tooth-natural Whitening Effect of Polyphosphate Containing Bleaching Products", I detected some MAJOR flaws, which determined my recommendation of “ACCEPT UNDER MAJOR REVIEW”. Below please find my detailed comments.

The authors performed a study investigated the tooth whitening effect of whitening products containing polyphosphates (OTC) compared to 10% carbamide peroxide "in vitro", using human teeth.

The research is written in a very confusing way and leaves several doubts throughout the text. The methodology needs to be significantly improved. The conclusion of the abstract does not match the findings obtained in the research.

My suggestions are bellow:

-        Abstract: Authors should explain what the abbreviations mean right in the abstract, because being the first time they appear in the text, it raises great doubts regarding the methodology and results, since the reader will not know what it is about: for example CIE L*, a*, b* and CIE-76-formula, that when reading the complete text, its meaning will only appear in the Material and Methods section, Color Measurement, after citing the acronyms in several paragraphs without explanation.

-        The conclusion described in the abstract does not match anything with the research carried out. The conclusion of the article is much more accurate in the full text.

-        It is not understandable how the samples were made, if the 8 human teeth were cut into smaller pieces, or the entire tooth was evaluated, as an n=20 per group appears at the end of the first paragraph of Material and Methods, which did not appear at any time.

-        In Results: in the item "Color difference" the authors cite a Table 4 that does not exist.

Author Response

Author's Reply to the Review Report (Reviewer 2)

After critically reviewing this Research Article titled "Biomimetic Tooth-natural Whitening Effect of Polyphosphate Containing Bleaching Products", I detected some MAJOR flaws, which determined my recommendation of “ACCEPT UNDER MAJOR REVIEW”. Below please find my detailed comments.

The research is written in a very confusing way and leaves several doubts throughout the text. The methodology needs to be significantly improved. The conclusion of the abstract does not match the findings obtained in the research.

Thank you for your comment, change was made

My suggestions are below:

-        Abstract: Authors should explain what the abbreviations mean right in the abstract, because being the first time they appear in the text, it raises great doubts regarding the methodology and results, since the reader will not know what it is about: for example CIE L*, a*, b* and CIE-76-formula, that when reading the complete text, its meaning will only appear in the Material and Methods section, Color Measurement, after citing the acronyms in several paragraphs without explanation.

Thank you for your comment.

-        The conclusion described in the abstract does not match anything with the research carried out. The conclusion of the article is much more accurate in the full text.

Thank you for your comment, change was made.

-        It is not understandable how the samples were made, if the 8 human teeth were cut into smaller pieces, or the entire tooth was evaluated, as an n=20 per group appears at the end of the first paragraph of Material and Methods, which did not appear at any time.

Thank you for your comment, change was made.

-        In Results: in the item "Color difference" the authors cite a Table 4 that does not exist.

Thank you for your comment, and you are correct, table was added by mistake. Change was made

Round 2

Reviewer 1 Report

Nice revision, my only objection is about the equation of DE, this is a scientific study there is an option of familiarity  about the methods of research. Since, you have the L, a, B parameters you should transform your results to the DE2000 and evaluate the results.

Author Response

Authors’ response to Reviewer 1: The authors thank Reviewer #1 for revising and providing opportunities to improve the submitted manuscript. The authors acknowledge the importance of reporting DE2000. However, the authors also understand that CIELAB is still the most commonly used method in color science publications and is still not substituted by DE2000. Therefore, the authors have addressed the limitation of not reporting in DE2000 in the current study, and certainly this will be seriously considered for subsequent studies. The discussion section has been modified accordingly to evidence the impact of the reviewer's comments on the authors' work. Also, in support of these comments, new references referring to the conversion and inclusion of both color systems were included.

Reviewer 2 Report

The modifications made are in line with my suggestions and those of other reviewers.

I was still a little in doubt about the making of the specimens, which could be better described.

Author Response

Authors’ response to Reviewer 2: Reviewer #2 is thanked for reviewing the manuscript in full and providing feedback to help the authors further improve this research findings' reporting.